# MicroBundleCompute: Automated segmentation, tracking, and analysis of subdomain deformation in cardiac microbundles

**Hiba Kobeissi[1,2], Javiera Jilberto[3], M. Çağatay Karakan[1,4,5], Xining Gao[5,6,7], Samuel J. DePalma[3], Shoshana L. Das[5,6,7], Lani Quach[3], Jonathan Urquia[8], Brendon M. Baker[3], Christopher S. Chen[5,7], David Nordsletten[3,9,10], Emma Lejeune[1,2]***

1 Department of Mechanical Engineering, Boston University, Boston, MA, United States of America, 2 Center for Multiscale and Translational Mechanobiology, Boston University, Boston, MA, United States of America, 3 Department of Biomedical Engineering, University of Michigan, Ann Arbor, MI, United States of America, 4 Photonics Center, Boston University, Boston, MA, United States of America, 5 Department of Biomedical Engineering, Boston University, Boston, MA, United States of America, 6 Harvard-MIT Program in Health Sciences and Technology, Institute for Medical Engineering and Science, Massachusetts Institute of Technology, Cambridge, MA, United States of America, 7 Wyss Institute for Biologically Inspired Engineering, Harvard University, Boston, MA, United States of America, 8 Department of Electrical and Computer Engineering, New York Institute of Technology, New York, NY, United States of America, 9 Department of Cardiac Surgery, University of Michigan, Ann Arbor, MI, United States of America, 10 Department of Biomedical Engineering, School of Imaging Sciences and Biomedical Engineering, King's Health Partners, King's College London, King's Health Partners, London, United Kingdom

* elejeune@bu.edu

**Data Availability Statement:** The data underlying the results presented in the study are available from https://datadryad.org/stash/dataset/doi:10.5061/dryad.5x69p8d8g. The code to reproduce the

## Abstract

Advancing human induced pluripotent stem cell derived cardiomyocyte (hiPSC-CM) technology will lead to significant progress ranging from disease modeling, to drug discovery, to regenerative tissue engineering. Yet, alongside these potential opportunities comes a critical challenge: attaining mature hiPSC-CM tissues. At present, there are multiple techniques to promote maturity of hiPSC-CMs including physical platforms and cell culture protocols. However, when it comes to making quantitative comparisons of functional behavior, there are limited options for reliably and reproducibly computing functional metrics that are suitable for direct cross-system comparison. In addition, the current standard functional metrics obtained from time-lapse images of cardiac microbundle contraction reported in the field (i.e., post forces, average tissue stress) do not take full advantage of the available information present in these data (i.e., full-field tissue displacements and strains). Thus, we present "MicroBundleCompute," a computational framework for automatic quantification of morphology-based mechanical metrics from movies of cardiac microbundles. Briefly, this computational framework offers tools for automatic tissue segmentation, tracking, and analysis of brightfield and phase contrast movies of beating cardiac microbundles. It is straightforward to implement, runs without user intervention, requires minimal input parameter setting selection, and is computationally inexpensive. In this paper, we describe the methods underlying this computational framework, show the results of our extensive validation studies, and demonstrate the utility of exploring heterogeneous tissue deformations and strains as

results in this work is available from https://github.com/HibaKob/SyntheticMicroBundle and https://github.com/HibaKob/MicroBundleCompute.

**Funding:** This work was supported by the CELL-MET Engineering Research Center National Science Foundation ERC ECC-1647837. EL acknowledges support from the American Heart Association Career Development Award 856354, XG acknowledges support from the National Science Foundation Graduate Research Fellowship, BMB and SJD acknowledge support from NSF Award 2033654, and SJD acknowledges support from NIH T32-DE007057 and NIH T32-HL125242. The funders had no role in study design, data collection and analysis, decision to publish, or preparation of the manuscript.

**Competing interests:** The authors have declared that no competing interests exist.

functional metrics. With this manuscript, we disseminate "MicroBundleCompute" as an open-source computational tool with the aim of making automated quantitative analysis of beating cardiac microbundles more accessible to the community.

## Introduction

Despite significant recent advances in cardiovascular disease prevention and diagnosis [1, 2], heart disease remains the leading cause of death among adults worldwide [3]. This is due, in part, to the fact that the native heart has a poor regenerative ability [4, 5], and thus damage to the heart muscle during an adverse medical event such as a myocardial infarction is irreversible [6]. Cardiac tissue engineering is a promising approach to address this unmet societal need [5]. In particular, cardiac tissue engineering with human induced pluripotent stem cell derived cardiomyocyte (hiPSC-CM) based technology [7] is a promising approach to disease modeling [8–11], drug discovery [5, 9, 11–13], and regenerative tissue engineering [14–16]. However, the development of viable hiPSC-CM technology is very much ongoing. In particular, one major challenge is that differentiated hiPSCs initially resemble fetal cardiomyocytes—they are morphologically and functionally different compared to adult cardiomyocytes [12]. Thus, developing technology to promote the maturation of hiPSC-CMs and, likewise, hiPSC-CM based tissue is an active area of research [16]. One impactful approach to promoting the maturation of hiPSC-CMs is the use of engineered tissue culture platform designs in both two [17, 18] and three dimensions [19–23] (Fig 1). Across these different platforms, there are multiple physical [17, 19, 22, 24], electrical [19, 20, 23–25], and chemical [26–28] knobs to tune to promote maturation and explore different physiological and pathological conditions. Even if we restrict our focus to microbundles (i.e., *aligned, electromechanically coupled, micro-scale cardiac tissue bundles formed with hydrogel materials suspended between pillars*), there is massive variability across different experimental setups [17, 19, 20, 22].

Driven by this diversity in experimental approaches and the rapid growth of the field, the mechanical behavior of cardiac microbundles is challenging to compare across studies. Fundamentally, this challenge is driven by multiple factors, ranging from the high volume of data collected with these testbeds [29], to challenges associated with reproducing results when software and data are not shared under open-source licenses, or when extracting quantities of interest from data requires significant manual processing. To date, there have been multiple non-destructive image-based methods for quantifying the contractile action of cardiac microbundles [21, 30–39], often inspired by related approaches to assessing the contractile behavior of cardiomyocytes [30, 40–50]. Broadly speaking, most of these tools can be grouped into four main categories: (1) edge detection systems [31, 32], (2) pillar tracking-based methods [33–37, 39], (3) inter-frame pixel disparity methods [21, 38], and (4) optical flow-based tracking [30, 39]. Each one of these methods has benefits and limitations that suit specific platforms, conditions, and research questions (some of these approaches are elaborated on in the "Materials and methods" Section). Despite this wide range of computational tools, many of which are available under open-source licenses [21, 33, 34, 38, 39, 51], few options can compare across multiple experimental testbeds and function on new datasets out of the box. In addition, the most popular approach to assessing microbundle contractile behavior—pillar tracking—does not necessarily capture the full richness of mechanical behavior in these systems.

Given this research landscape, there is a clear need for an open-source computational tool to extract functional metrics from time-lapse images of microbundle contraction. To address

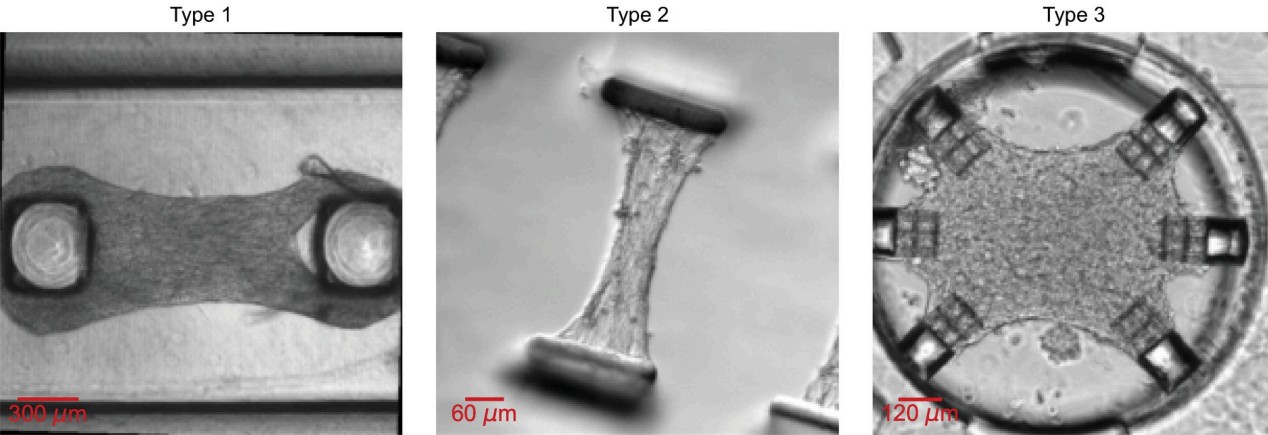

**Fig 1. Examples of 3 microbundle experimental testbeds of different types.** We briefly note here that "Type 1" and "Type 3" represent 3D culture platforms while "Type 2" portrays a significantly thinner (almost 2D) platform. We elaborate more on each type in the "Experimental data" Section and in S2 Appendix.

this need, we have developed the "MicroBundleCompute" computational framework that we will present in this paper. In brief, "MicroBundleCompute" is disseminated as a Python package and is based on the Lucas-Kanade optical flow algorithm [52] for computing full-field displacements, subdomain-averaged strains, and displacement and strain derived metrics. In this manuscript, we introduce the methods underlying the "MicroBundleCompute" framework, discuss our approach to validating the pipeline primarily via realistic synthetic data, and then show the results of implementing "MicroBundleCompute" on multiple testbeds, shown in Fig 1. We not only portray the efficacy of our approach, but also show that examining full-field tissue deformation consistently reveals heterogeneous contractile behavior throughout the domain. Looking forward, our goal is to use this work as a starting point to move beyond previously developed ad hoc approaches to analyzing these data [53], and establish a computational foundation for performing cardiac microbundle assessment and quality control at scale.

The remainder of this document is organized as follows. In the "Materials and methods" Section, we describe our methods to generate synthetic movies of beating cardiac microbundles with known ground truth to validate our computational pipeline, the diverse set of real examples on which we test the software, and finally, our code pipeline along with the main output metrics. Then in the "Results and discussion" Section, we present the main findings of our code validation on one synthetic example and show sample outputs on a few real movies. Finally, in the "Conclusion" Section, we share our final thoughts regarding "MicroBundleCompute." Along with this document, we provide three supplementary documents where we explain in more details our validation pipeline (S1 Appendix), share our complete set of real examples (S2 Appendix), and finally provide more information on basic pillar tracking (S3 Appendix), an additional feature of our computational tool. Overall, the intention of this manuscript is to outline our method and approach to making a tool of broad utility for the cardiac tissue engineering research community.

## Materials and methods

In a broader sense, capturing full-field soft tissue deformation is critical for a number of research applications ranging from inverse material characterization [54–57], to high-fidelity biomechanical modeling of in vivo mechanisms [58–60], and patient-specific modeling and

procedure planning [61, 62]. And recently, there has been keen interest in implementing image registration-based techniques widely used in the field of computer vision [54, 63] for full-field measurements, including digital image correlation (DIC) [64–68], 3D-DIC [69, 70], digital volume correlation (DVC) [71–74], and optical flow algorithms [75–77].

With regards to applications to cardiac microbundles in specific, available image-based techniques for computing the contractile behavior of microbundles, as categorized in the "Introduction" into 4 broad approaches, focus mainly on quantifying temporal profiles for the entire construct and as such, provide averaged results that lump the spatially heterogeneous tissue behavior into a single value per frame (3 of the 4 approaches). However, tools developed based on optical flow ($4^{th}$ approach) can extract full-field, as well as directional outputs, such as directional displacement fields and strains.

To elaborate more, edge-detection approaches rely on quantifying the microbundle shape change between a relaxed (reference) state and a contracted (deformed) state. For example, in Ronaldson-Bouchard et al. [31], tissue contractility was calculated by tracking the change in the tissue area while previously, in Hansen et al. [32], the difference between the ends of the tissue was used to measure contraction. Yet, these tools were custom-developed and are not readily available online for the broader research community. As for pillar tracking-based methods, there are currently a number of available tools [33, 34, 39] out of the identified implementations that were developed and kept in-house [35–37]. In these approaches, the pillar or cantilever head deflection is estimated to generate contraction waveforms and extract contraction kinetics including contraction frequency, force, as well as the time to achieve 10%, 50%, or 90% of the peak contraction or relaxation.

Available tissue tracking methods via pixel intensity disparity have been implemented based on different approaches. For example, in "MUSCLEMOTION" [51], which is offered as an ImageJ [78] plugin, the absolute difference in pixel intensity between a reference frame and a frame of interest is calculated, whereas the MATLAB-based [79] "CardiacContractileMotion" [21] identifies the tissue region in a relaxed baseline state and tracks, within this region of interest, changes in pixel motion with time. Another MATLAB-based [79] tool, "ContractQuant" [38], which was specifically developed for implementation with micron-scale 2D cardiac muscle bundles, uses cross-correlation to track pixel features and find the best match for a specified region of interest across consecutive frames. Outputs from these software are in general similar to those extracted with pillar tracking approaches and include contraction and relaxation profiles and velocities, as well as contraction and relaxation times.

Overall, these approaches are suitable when only averaged values, such as the mean value and the time rate or velocity of tissue shortening, shrink or contraction, are enough. However, tools developed based on optical flow can provide richer outputs. For example, Huebsch et al. [30] implemented block-matching optical flow [80] methods in MATLAB [79] to estimate absolute as well as directional full-field contractility. And very recently, a particle image velocimetry toolbox in MATLAB [79] was utilized in [39] to calculate displacement vectors of subregions or patches identified within a manually selected region of interest in the "reference" frame using a cross-correlation approach. And from these displacement fields, strain maps were subsequently derived. To the best of our knowledge, these methods appear to be robust and relatively versatile, yet at present they lack automation.

Within this scope, we present here our high-throughput optical flow-based computational framework to extract full-field deformation metrics from lab-grown cardiac microbundles. In the Sections that follow, we describe both the data used for testing and validating the "MicroBundleCompute" software, and the details of our computational methods. First, in the "Data" Section, we introduce the two general categories of cardiac microbundle data that we have used in developing the software: synthetic data with a known ground truth and experimental

data. Next, in the "Code" Section, we explain the details of the code pipeline and describe core functionalities and output metrics.

## Data

To validate our pipeline, we first invest significant effort into creating labeled data with a known ground truth. In the "Synthetic data generation" Section, we outline our process for synthetic data generation. As a brief note, this is complemented with additional information in S1 Appendix where we show comparisons to another form of synthetic data and manually labeled experimental data. In the "Experimental data" Section, we then show three distinct classes of microbundle experimental testbeds that we will use to showcase the function and versatility of our software.

**Synthetic data generation.** Here, we describe the steps to generate realistic synthetic brightfield movies of beating cardiac microbundles based on examples from "Type 1," as described in the "Experimental data" Section. In Fig 2, we summarize the main steps of this pipeline. For each synthetic data example, we begin with a frame from an experimental movie (Fig 2a). As the first step of this pipeline, we manually trace the tissue region in a relaxed valley frame, and use the traced region to obtain both tissue geometry and image texture. From the traced region, we extract the coordinates of the external contour to generate Finite Element (FE) simulation geometry, and isolate the tissue texture that will be warped following the FE simulation results (Fig 2b).

To inform our FE simulations, we first generate a simplified three-dimensional microbundle geometry based on the contour coordinates of a mask extracted from the single representative valley frame (Fig 2c). Specifically, we extrude the 2D surface created by connecting these contour coordinates to a thickness of $400\mu$m, a reasonable microbundle thickness given our target experimental setups [22]. To approximate the pillars, we implement the geometry and dimensions detailed in [22], which matches one of the main platforms used in our experimental dataset ("Type 1"). To create the FE mesh, we use Gmsh 4.10.5 [81], where the final mesh consists of 205, 524 tetrahedral elements which was deemed sufficient for our purpose, following a mesh refinement study. We provide a detailed schematic of the three-dimensional mesh geometry in S1 Appendix.

In Fig 2d, we briefly summarize the main components of the FE model as implemented in FEniCS 2019.1.0 [82, 83]. Following popular recent work in the field of soft tissue biomechanics [84–86], we model the cardiac tissue as a nearly-incompressible transversely isotropic hyperelastic material where deformation is driven by periodic activation [84, 86, 87]. Of note,

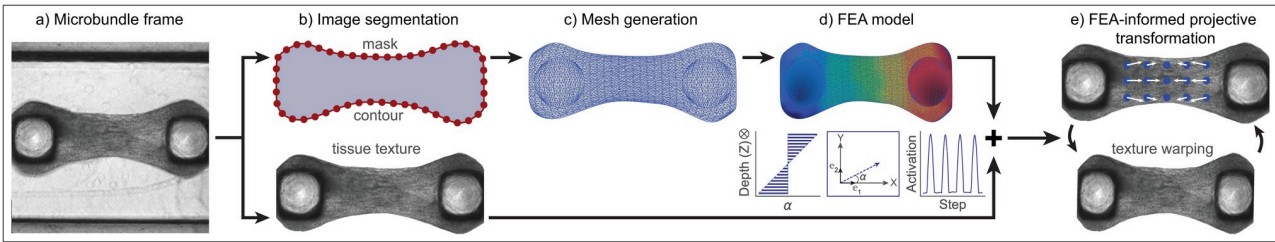

**Fig 2. Schematic representation of the synthetic data generation pipeline.** Illustrations of the main elements of the synthetic data generation pipeline in order of implementation: (a) an image of a microbundle movie frame; (b) a mask of the microbundle, extracted contour coordinates, and a segmented tissue texture; (c) a three-dimensional mesh for the FE model; (d) main variables within the FE simulations in order from left to right: profile of a linearly variable fiber direction with respect to depth, an illustration of the tissue depth direction, a uniform time series activation; (e) extracted surface displacement results and a warped image texture based on an estimated projective transformation.

we create synthetic data with heterogeneous deformation fields (specifically, (1) a fully actively contracting tissue domain, (2) a passive circular inclusion at the center of the actively contracting tissue), and we vary the direction of contractile alignment (1) through the thickness or depth of the tissue or (2) along its length. We model the poly(dimethylsiloxane) (PDMS) pillars as passive Neo-Hookean material, and treat the interface between the cardiac tissue and the pillars as perfectly bonded. For each FE simulation, we extract, with respect to time, the $X$, $Y$, and $Z$ positions of the mesh cell centers at the top surface of the microbundle for each step and save the results as text files. This simplified FE model serves a single purpose: to computationally generate synthetic data of realistically beating microbundles with known ground truth deformation. Then, following the schematic illustration in Fig 2e, we estimate a projective transformation based on the initial and deformed positions of the mesh cell centers and warp the image texture accordingly using the `warp` transform function in the scikit-image 0.19.3 Python library [88]. To enable a heterogeneous transformation, we subdivide the image domain and perform subdomain specific warping. Additional details are available in S1 Appendix.

Overall, our main synthetic dataset consists of 60 generated movies of beating experimentally derived image textures. To obtain these 60 examples, we use 15 different base texture images extracted from 5 experimental movies of "Type 1" as described in the "Experimental data" Section. We then deform these extracted textures with FE results obtained from 4 different FE simulations run under the variable conditions specified above. To perform quantitative evaluation, we extract a $90 \times 90$ pixel region from each domain center and make all direct comparisons based on this domain.

In addition, we perform additional validation against a single computationally generated synthetic example of "Type 2" data based on a more sophisticated tissue-specific FE model described in detail in Jilberto et al. [89] and S1 Appendix. For more information on implementing the Finite Element model, image warping, and the addition of Perlin noise [90], we refer the reader to S1 Appendix. We also make the entire synthetic dataset prior to the addition of Perlin noise available along with all the Python code and files that are necessary to re-generate our dataset available on GitHub (https://github.com/HibaKob/SyntheticMicroBundle).

**Experimental data.** Our experimental dataset can be systematically categorized into 3 different types (Fig 1). "Type 1" includes movies obtained from standard experimental microbundle strain gauge devices [91–93]. We refer to data collected from non-standard platforms termed FibroTUGs [94] as "Type 2" data. As for "Type 3," they represent data obtained from a highly versatile experimental platform [19, 20] and as such, include the most diverse examples in this collection.

Specifically, "Type 1" examples were prepared as previously detailed in [53]. Briefly, PDMS (Dow Silicones Corporation, Midland, MI) microbundle devices were first cast from 3D printed molds (Protolabs). Each device contains 6 wells, each with two pillars with rectangular cross sections and spherical caps, where the cardiac microbundles are seeded. Up to 2 days before seeding, the devices were sequentially treated with 0.01% poly-l-lysine (ScienCell) and then with 0.1% glutaraldehyde (EMS) to promote cell attachment to the caps. On the day of seeding, devices were cleaned with 70% ethanol and ultraviolet (UV) sterilized. Next, the device wells were incubated with a small volume of 2% Pluronic F-127 (Sigma) to prevent cell attachment at the base of the well. hiPSC-CMs, differentiated and purified as described by Lian et al. [95], were seeded with human ventricular cardiac fibroblasts in a Matrigel (Corning) and fibrin (Sigma) extracellular matrix (ECM) solution. Microbundles were maintained in growth medium, with replacement every other day. Time-lapse videos of tissue contractions were acquired 5–7 days after seeding at 30 Hz using a 4× objective on a Nikon Eclipse Ti

(Nikon Instruments Inc.) with an Evolve EMCCD camera (Photometrics), while maintaining a temperature of 37˚C and 5% $CO_2$.

As for the second type, FibroTUG microbundles were fabricated as described previously [17, 94]. First, arrays of PDMS cantilevers were fabricated by soft lithography as detailed in [94]. Then, fiber matrices, suspended between pairs of these cantilevers, were generated by selective photo-crosslinking of electrospun dextran vinyl sulfone (DVS) fiber matrices deposited onto the microfabricated PDMS cantilevers [17, 94]. Matrix and cantilever stiffnesses were tuned by adjusting photoinitiator concentrations and cantilever height, respectively, while matrix alignment was controlled by altering the translation speed of collection substrates during fiber deposition [17, 94]. Following functionalization of the electrospun fiber matrices with cell adhesive cRGD peptides, iPSC-CMs, differentiated and purified [17], were patterned onto matrices using microfabricated seeding masks cast from 3D-printed molds. Finally, time-lapse videos of the microbundle's spontaneous contractions were acquired at $\sim$ 65 Hz on Zeiss LSM800 equipped with an Axiocam 503 camera while maintaining a temperature of 37˚C and 5% $CO_2$.

Examples from "Type 3" were generated using the protocol previously described in [19, 20]. In brief, a combination of soft lithography and two-photon direct laser writing (DLW, Nanoscribe Photonic Professional GT+) was used to fabricate the seeding platforms. The process involves printing negative master molds using DLW, casting PDMS onto the molds, followed by sandwiching, curing and demolding. This results in 0.5—0.6 mm-thick PDMS devices with embedded microfluidic channels and deformable seeding wells. As a final step, cage-like microstructures were printed using DLW on the sides of the wells of the demolded PDMS devices to facilitate cell attachment. After device fabrication, differentiated hiPSC-CMs as per the procedures described in [19], were seeded into the wells with human mesenchymal stem cells in a collagen ECM solution, with the growth medium changed every other day. Time-lapse videos of the tissue contractions were acquired 4–9 days after seeding at 30 Hz using 4× or 10× objectives on a Nikon Eclipse Ti (Nikon Instruments Inc.) with an Evolve EMCCD camera (Photometrics) equipped with a temperature and $CO_2$ equilibrated environmental chamber.

In total, we include in this framework 24 real experimental data, 11 examples from "Type 1," 7 from "Type 2," and 6 from "Type 3." This diverse pool of examples allows us to not only demonstrate the adaptability of our computational pipeline to different input examples, but also gain valuable insight about the heterogeneous contractile action of cardiac tissue by extracting and observing relevant mechanical metrics, such as full-field displacements, subdomain-averaged strains, and displacement and strain-derived outputs, as shown in the "Experimental data examples" Section and in S2 Appendix. We note that details about each specific experimental example are provided in S2 Appendix as well as on Dryad [96] (https://doi.org/10.5061/dryad.5x69p8d8g) where the whole dataset, "Microbundle Time-lapse Dataset," is made available.

## Code

In this Section, we describe the main working components of our "MicroBundleCompute" software for the automatic analysis of deformation in brightfield and phase contrast movies of cardiac microbundles. Because our goal is to implement an approach with simplicity, versatility, and adaptability in mind, our pipeline is structured with four modular components: (1) image pre-processing and mask creation, (2) deformation tracking, (3) post-processing (e.g., rotation, interpolation, strain analysis), and (4) visualization. In Fig 3, we provide a graphical summary of the major functionalities included in this pipeline and the computational

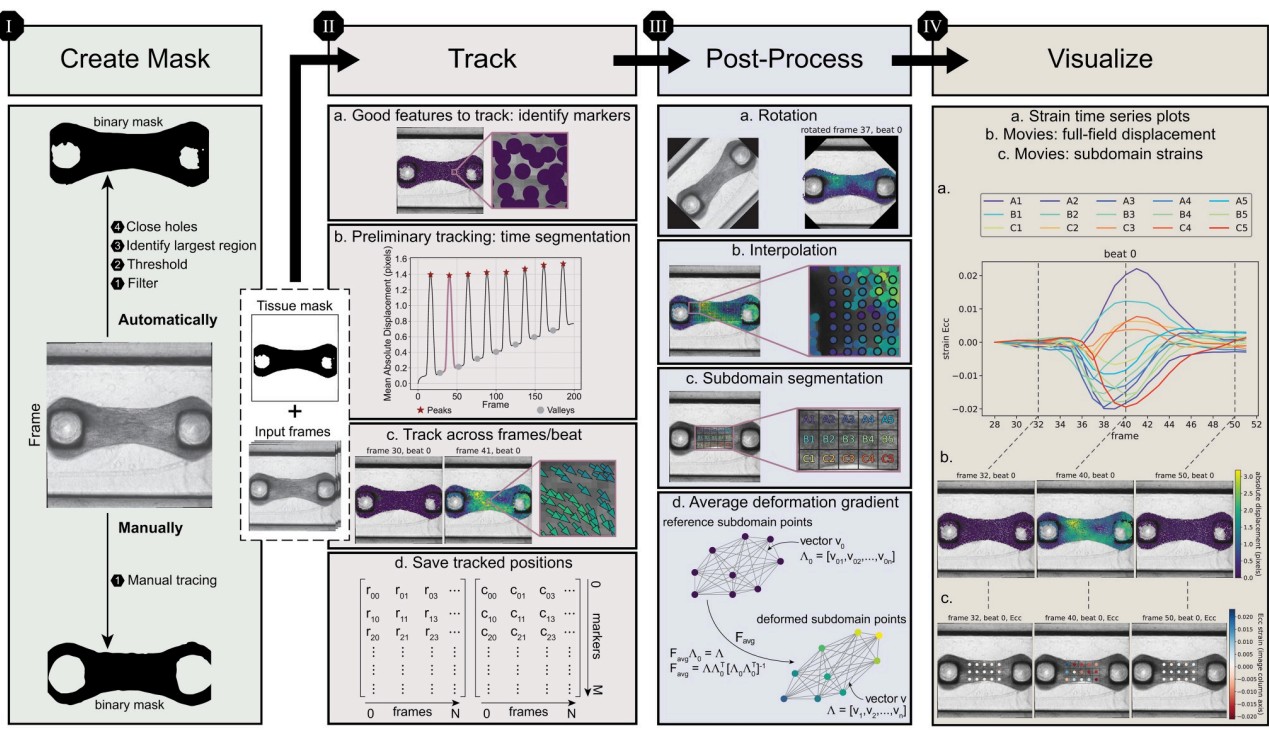

**Fig 3. Schematic illustration of the "MicroBundleCompute" computational framework.** Two main inputs are required: I) a binary mask generated either automatically using the software or manually and consecutive movie frames of the beating microbundle. II) For tracking, (a) marker points are identified on the first frame and (b) tracked across all frames to identify individual beats and perform time segmentation. (c) This allows us to perform the analysis per beat and correct for the observed drift as discussed in the "Temporal segmentation" Section. (d) Finally, we save the row and column positions of the tracked markers per a single beat and use these saved outputs to compute full-field displacements and derive strain results. III) Post-processing functionalities include (a) rotation of the images and tracking results, (b) interpolation of the results at query points, and (c) segmentation into subdomains for which (d) average deformation gradients and subsequently, subdomain-averaged strains are calculated. IV) Finally, to visualize the results, the software outputs (a) time series plots per beat and movies of (b) full-field and (c) subdomain-averaged results.

workflow. As a brief note, the software GitHub repository (https://github.com/HibaKob/MicroBundleCompute) contains instructions on how to install and run the code, detailed explanations of each main function, and a more thorough description of the formatting of output files.

The implementation of these methods is divided across four Python files: `create_tissue_mask`, `image_analysis`, `strain_analysis`, and `optional_preprocessing`. The bulk of core functionality is contained in `image_analysis`, and all functions of the code are designed to be modular when possible such that they can be replaced in the future if a need arises. In addition, many of the specific post-processing steps are optional, and adaptation to compute additional quantities of interest should be straightforward. As we describe our pipeline, we will specify which Python file a given function is located in. Essential functions included in our pipeline are as follows, following the order illustrated in Fig 3.

**Tracking region mask generation.** Within `create_tissue_mask`, we provide two different options for creating a binary mask of the tissue region: manual or automatic (Fig 3I). At present, we provide 3 basic segmentation functions for automatic microbundle mask creation: 1) a straightforward threshold-based mask, 2) a threshold-based mask that is applied after a Sobel filter, and 3) a threshold-based mask that is applied to either the minimum or the

maximum (specified by the user as an input) of all movie frames. We implement our tissue segmentation pipeline mainly based on the `threshold_otsu` and `sobel` functions provided within the `filters` module in scikit-image 0.19.3 Python library [88].

In addition to programming alternative automated mask generation functions, software users can provide an externally generated tissue mask (e.g., a manually generated mask) by including a file named "tissue_mask.txt" where the mask is a two-dimensional array in which the tissue domain is denoted by "1" and the background domain is denoted by "0" to allow for the analysis of domains that may fall outside the original scope of this endeavor.

**Sparse optical flow algorithm for tracking.**   Within `image_analysis`, we provide all of the essential functions to automatically run the tracking algorithm illustrated in Fig 3II. Our tracking pipeline is built on OpenCV's [97] pyramidal implementation of the Lucas-Kanade sparse optical flow algorithm [52] where we identify markers as Shi-Tomasi "good features to track" corner points [98] that fall within the specified tissue mask. In brief, these corner points are points where a slight shift in location leads to "large" changes in intensity along both the horizontal and vertical axes. There are two key parameters required to tune OpenCV's `goodFeaturesToTrack` function: `qualityLevel`, a minimum score to measure if a feature can be tracked well, and `minDistance`, the minimum permitted Euclidean distance between two identified corners. We initialize `minDistance` = 3 and `qualityLevel` = 0.1. Then, we iteratively decrease `qualityLevel` until `coverage` > 40, where `coverage` is defined as the average number of pixels associated with each tracking point, for up to 15 iterations. As for `minDistance`, we define a `local_coverage` measure, which is the `coverage` computed on 20 × 20 pixel subdivisions. The `minDistance` is automatically incremented by 1 as long as the largest 3 `local_coverage` values are less than or equal to 50 and the number of iterations does not exceed 2. We note that we adjust `qualityLevel` and `minDistance` simultaneously.

For running OpenCV's [97] Lucas-Kanade optical flow [52] function (`calcOpticalFlowPyrLK`), we automatically tune the parameter `winSize`, which dictates the size of the integration window. Crucially, the window size in both horizontal and vertical directions, $w_x$ and $w_y$, should be larger than the maximum tracked pixel motion between frames. To specify `winSize`, we adopt a pragmatic approach where we initialize `winSize` = 5, perform a preliminary tracking step, calculate the maximum absolute displacement, and compare its magnitude to the initial window size. If the calculated displacement is larger, we increase `winSize` by 5, and continue to iterate until the condition is met or the number of iterations exceeds 15. Critically, keeping `winSize` from being larger than necessary reduces error during tracking. In the remainder of this Section, we will describe our methods for leveraging this basic sparse optical flow algorithm to effectively and automatically analyze microbundle domains.

**Temporal segmentation.**   After automatic identification of features and tracking algorithm parameters, we run a preliminary tracking step. Representative results of preliminary tracking are shown in Fig 3II.b as a plot of mean absolute displacement vs. frame number. We use these preliminary tracking results to perform temporal segmentation where we delineate individual tissue beats. To accomplish this, we use the SciPy signal processing library function `find_peaks` [99]. The `find_peaks` input parameters, `distance`, the minimum horizontal distance between neighbouring peaks, and `prominence`, minimum prominence for a perturbation to be recognized as a peak, are identified automatically. Specifically, we initialize `distance` and `prominence` to values of 20 and 0.1 respectively. Then `distance` is updated to take a value equal to 1.5× the horizontal distance separating two consecutive intersection points between the time series and its mean. For the `prominence` parameter, we keep it constant at a value of 0.1, which we found to be suitable for all the example videos that

we have analyzed to date. After peaks are identified, we define valleys as the midpoints of two consecutive peaks. As such, to be able to identify a pair of valleys, or in other words a single beat, a minimum of 3 beats should be present in any movie to enable automatic analysis and an accurate approximation of beat period. Temporal segmentation into individual beats is then performed based on the temporal location of each valley.

Beyond the determination of microbundle time-related properties (e.g., beat period), temporal segmentation is an essential part of our pipeline as we use it to work around the tracked feature drift observed over the duration of the movie (see Fig 3II.b for an illustration of drift).

After identifying individual beats, we split, based on the first and last beat frame numbers, the main two arrays storing column (horizontal) and row (vertical) locations of marker points obtained during preliminary tracking into multiple arrays corresponding to each segmented beat. Likewise, instead of frame 0 being the reference configuration for the whole tracking duration, the fiducial marker positions in the first frame of each beat become the baseline for all future output calculations within the beat.

**Optional rotation and interpolation.** After the tracking step is complete, we include two optional features for post-processing: sample rotation and fiducial marker displacement interpolation. First, we include an option to rotate both the images and the tracking results based on a specified center of rotation and desired horizontal axis vector. The center of rotation and desired horizontal axis vector can be either specified manually or identified automatically based on the geometry of the tissue mask. As a brief note, rotation is performed after tracking as the process involves interpolation which can lead to loss of image resolution. Also, we automatically rotate the tissue domain before performing strain subdomain calculations to match the global row (vertical) and column (horizontal) coordinate system. The second optional feature, interpolation, allows the user to interpolate the tracking results returned at the automatically identified fiducial marker points to user-specified locations, on a structured grid for example. This step can be performed after tracking and optional rotation, and will output the interpolated displacement fields to specified sampling points for either visualization or downstream analysis.

**Subdomain spatial segmentation.** In contrast to standard approaches to analyzing cardiac microbundles [51], our approach is unique in that we compute full-field quantities of interest over the tissue domain. In order to better analyze these full-field results and reliably post-process displacement fields to compute strain, we perform spatial segmentation to define tissue subdomains over which we can report average strain quantities [100, 101]. This subdomain spatial segmentation is implemented in the `strain_analysis` file within "MicroBundleCompute." We provide two options to specify the subdomain extents defined as a rectangle: 1) automatic subdomain generation via clipping the input tissue mask or 2) manually providing subdomain extent coordinates. Given rectangular subdomain extents, we then delineate individual subdomain tiles by specifying either the target number of tiles in each column and row or by specifying the target tile dimensions in pixels. Representative subdomain segmentation results are illustrated in Fig 3III.c.

**Strain computation.** With these defined subdomain regions, we then compute the average deformation gradient $\mathbf{F_{avg}}$ of each subdomain and use it to compute relevant strain metrics. As stated previously, we compute average subdomain strain rather than full-field strain due to: (1) the desire to reduce the influence of imaging artifacts and noise, and (2) increased ease of comparison between samples. Within each subdomain, we define the standard continuum mechanics deformation gradient $\mathbf{F}$ as follows:

$$\mathbf{F}d\mathbf{X} = d\mathbf{x} \tag{1}$$

where $\mathbf{F}$ maps a vector $d\mathbf{X}$ in the initial or reference configuration to its deformed configuration $d\mathbf{x}$ [102]. To apply this within the context of our tracking pipeline, we define a set of $n$ vectors, $\mathbf{\Lambda_0}$, that connect each potential pair of fiducial markers that lie within the extents of each subdomain in the reference configuration. We define the reference configuration with respect to each cardiac tissue beat as the first frame in the segmented beat frame. Thus, $\mathbf{\Lambda_0}$ is defined in frames that represent the most relaxed tissue state. We then compute $\mathbf{\Lambda}$ following the same structure as $\mathbf{\Lambda_0}$ for each subsequent movie frame, where the updated fiducial marker positions capture the subdomain deformed configuration. With this definition, we can set up the over-determined system of equations:

$$\mathbf{F_{avg}\Lambda_0} = \mathbf{\Lambda} \quad \text{where} \quad \mathbf{\Lambda_0} = [\mathbf{v_{01}}, \mathbf{v_{02}}, \ldots, \mathbf{v_{0n}}] \quad \text{and} \quad \mathbf{\Lambda} = [\mathbf{v_1}, \mathbf{v_2}, \ldots, \mathbf{v_n}] \tag{2}$$

where $\mathbf{F_{avg}}$ is a $2 \times 2$ matrix, and $\mathbf{\Lambda_0}$ and $\mathbf{\Lambda}$ are $2 \times n$ matrices of vectors in the initial (reference) and current (deformed) configurations, respectively. Of note, when the initial and current frames are identical, $\mathbf{\Lambda_0} = \mathbf{\Lambda}$, $\mathbf{F_{avg}} = \mathbf{I}$, a $2 \times 2$ identity matrix. To solve this over-determined system, we can use the normal equation to find the best fit average deformation gradient as:

$$\mathbf{F_{avg}} = \mathbf{\Lambda\Lambda_0^T}[\mathbf{\Lambda_0\Lambda_0^T}]^{-1}. \tag{3}$$

We schematically illustrate our method to compute the mean deformation gradient in Fig 3III.d. With the computed $\mathbf{F_{avg}}$, finding the average Green-Lagrange strain tensor is straightforward:

$$\mathbf{E_{avg}} = \frac{1}{2}\left(\mathbf{C_{avg}} - \mathbf{I}\right) \quad \text{where} \quad \mathbf{C_{avg}} = \mathbf{F_{avg}^T F_{avg}} \tag{4}$$

and we can then compute strain on a per subdomain per beat basis to obtain subdomain time series results for $\mathbf{F_{avg}}$ and subsequently $\mathbf{E_{avg}}$.

**Data structure preparation.** Tracking all identified fiducial markers for an extended number of frames produces a large quantity of output data for each movie. Thus, we selectively save output results such that they are both comprehensible and easily accessible for downstream data analysis. First, we save information regarding the column (horizontal) and row (vertical) positions of the tracked marker points per beat. Specifically, we store one row-position text file and one col-position text file for each beat formatted as a $M \times N$ array where $M$ is the number of markers that were tracked and $N$ is the number of frames in the beat. Similarly, we output mean deformation gradient results as text files saving the column and row positions of the center for each subdomain and the 4 components of the $2 \times 2$ mean deformation gradient per subdomain per beat.

**Data analysis, key metrics, and visualization.** In Fig 3IV, we show key output metrics and their visualizations. In brief, we provide tools to visualize full-field displacement and average subdomain strain, and provide key quantities of interest such as maximum strain, beat period, and synchrony. In all cases, we build on the popular matplotlib package [103] for producing all visualizations. In addition to the metrics directly enabled by our novel pipeline, we provide other commonly pursued relevant outputs including beat frequency, beat mean amplitude, and tissue width at the domain center. To convert the numerical outputs from dimensions of pixels and frame numbers to physical units, the user can specify: 1) frames per second and 2) the length scale in units of $\mu$m/pixel.

**Quality checking and rejection of unsuitable examples.** In our extensive experience testing our code on different synthetic and real examples, we have identified three main instances that negatively influence the fidelity of our outputs and decrease our confidence in analysis results:

1. blurred input movie frames that prevent effective identification of corner points for the tracking described in the "Sparse optical flow algorithm for tracking" Section.

2. movies that start from a contracted tissue position that confounds the temporal segmentation step described in the "Temporal segmentation" Section.

3. movies where all displacement is on *sub*-pixel length scales lead to large relative errors given our choice of tracking algorithm.

The specific influence of these conditions are explored in S1 Appendix. To address these conditions, we provide both warnings in the code when we detect that these conditions arise and functions to correct these scenarios (e.g., removing initial movie frames to address case 2).

**Note on pillar tracking.** In the broader cardiac microbundle and microtissue literature [33, 39], pillar force and tissue stress are two commonly computed metrics. Broadly speaking, pillar force is computed by tracking pillar displacement and converting displacement to force via cantilever beam equations where the pillars are treated as beams with known elastic modulus and geometry [104]. Stress is then computed from pillar force by dividing force by tissue cross sectional area where tissue width is measured from the in-plane images [34] and tissue depth is assumed based on typical values observed via three dimensional imaging modalities [105]. We are able to adapt our framework to readily track pillar displacement by simply specifying pillar regions as the area to track rather than the tissue domain. Due to lack of novelty, we do not emphasize this functionality in this paper. However, we do provide additional details on this topic in S3 Appendix.

## Results and discussion

In this Section, we show a summary of our main results. In the "Synthetic data examples" Section, we present representative validation examples from validating our pipeline on an example from the synthetic dataset of "Type 1" with known ground truth behavior, and in the "Experimental data examples" Section, we share examples of implementing our computational framework on 3 different experimental platforms (i.e., 3 different data types as explained in the "Experimental data" Section). As a brief note, we provide a more comprehensive set of results in S1 Appendix on validation, S2 Appendix on additional real data examples, and S3 Appendix on pillar tracking.

### Synthetic data examples

Here, we briefly summarize the results of our validation studies. In Fig 4, we show the performance of our pipeline on a single representative synthetic example ST_1. In S1 Appendix, we provide a comprehensive summary of all 416 synthetic validation examples based on data of "Type 1" as well as a single synthetic example of "Type 2" data. In our validation studies, we compare "MicroBundleCompute" displacement and strain outputs against the known ground truth for synthetic data as originally generated (without noise) and for the same examples with added Perlin noise of different magnitude ratios and octaves. As a brief note, other types of noise such as shot noise [106] would be appropriate alternatives to Perlin noise for the purpose of this investigation. In preliminary studies, we found that Perlin noise with a range of magnitudes and octaves was a sufficiently general choice to provide a robust challenge for our framework. The code required to reproduce all synthetic data examples of "Type 1" can be found on GitHub (https://github.com/HibaKob/SyntheticMicroBundle).

In Fig 4a, we specify the synthetic data domain for which the validation studies were performed, a $90 \times 90$ domain warped with FEA-informed displacements as briefly described in the "Synthetic data generation" Section and presented in more details in S1 Appendix. In Fig

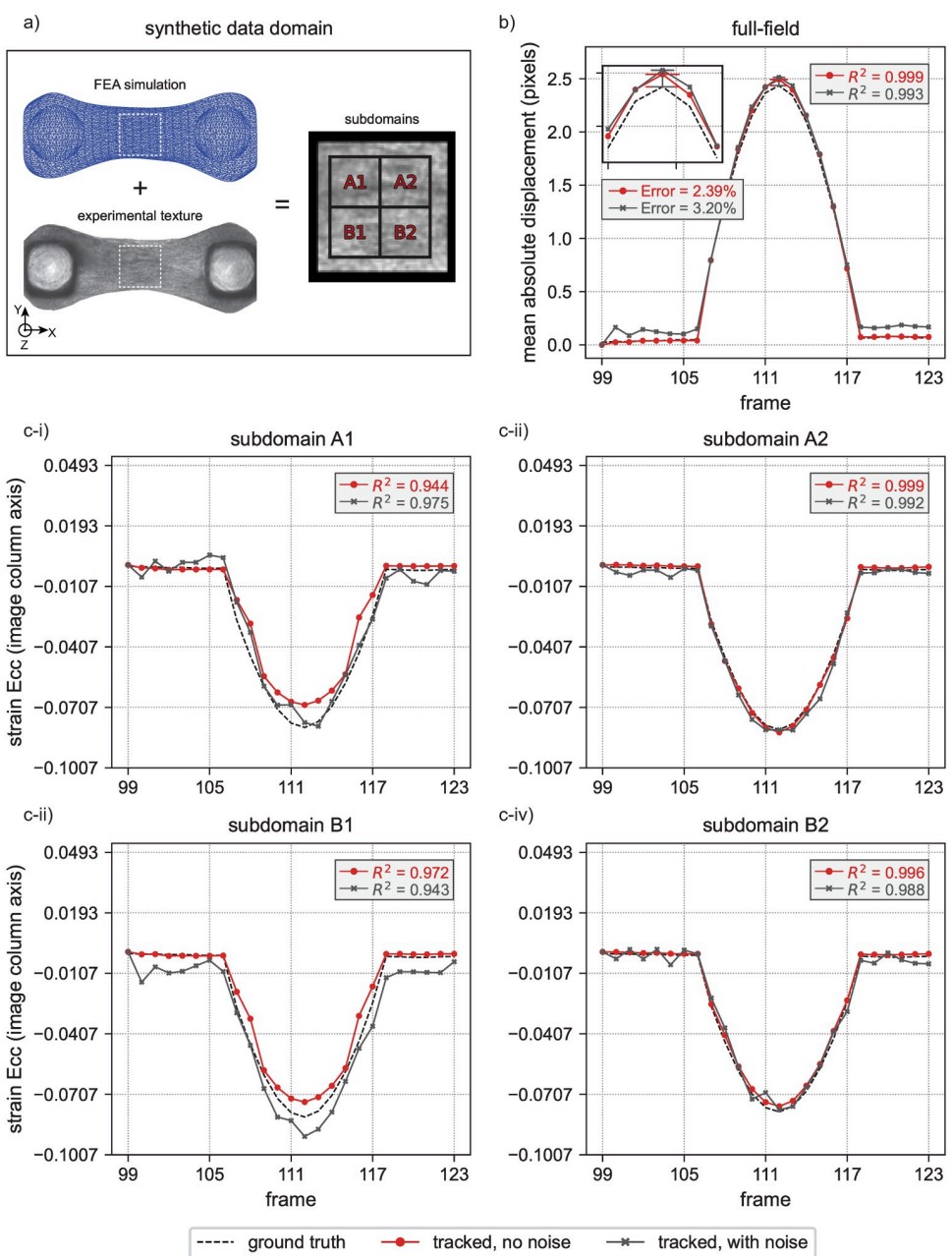

**Fig 4. Validation against a single synthetic example.** Validation results against synthetic example ST_1 in its original (no noise) state as well as after the addition of Perlin noise (magnitude ratio of 12% and octaves of 40): (a) a schematic representation of the synthetic data domain and subdomain divisions for strain calculations; (b) tracked and FEA-extracted (ground truth) mean absolute displacement of the full-field data for a single beat (beat 4), where maximum error for synthetic data with noise is observed; (c-i)–(c-iv) tracked and FEA-extracted (ground truth) subdomain-averaged $E_{cc}$ strain for beat 4 for each subdomain. Overall, the $R^2$ values indicate relatively good agreement between the tracking output and ground truth data. For more information on validation with synthetic data, refer to S1 Appendix.

4b, we plot the mean absolute displacement with respect to frame number for beat 4, the beat with the maximum error at the peak displacement (full contraction) for the synthetic example with Perlin noise of magnitude ratio of 12% and octaves of 40. We specifically highlight these Perlin noise parameters because they mimic, to a great extent, noise artifacts found in the real

experimental data. A direct comparison between both tracked and ground truth mean absolute displacements per beat revealed that the percentage error at peak displacement is less than 2.5% for the example shown in Fig 4b, with the addition of Perlin noise slightly raising this error to 3.2%. In addition, we assessed the ability of our computational framework to "predict" the mean absolute displacement by calculating the coefficient of determination ($R^2$) and found good agreement. Overall, considering all validation examples in S1 Appendix prior to the addition of Perlin noise, peak mean displacement errors were found to be less than 7% for *super*-pixel displacements (Table S1 _1 and Fig S1_5 in S1 Appendix) whereas *sub*-pixel displacements had higher errors that reach up to 15% (Table S1_1 and Fig S1_6 in S1 Appendix) with a minimum $R^2$ value of 0.939. Furthermore, it is evident that the addition of Perlin noise produces higher shifts within the tracked outputs, with higher errors reported for higher magnitude ratios and lower octaves (Figs S1_23 and S1_24 in S1 Appendix). Specifically, as shown in Table S1_4 and Fig S1_23 in S1 Appendix, the percentage error at the peak mean absolute displacement increases to around 14.4% for *super*-pixel displacement examples. For the *sub*-pixel displacement examples, mean absolute displacement errors become unreasonably high in the majority of the synthetic examples and are on the order of $10^3 - 10^4$ (Fig S1_24 in S1 Appendix). And, in some of these *sub*-pixel displacement examples where the added Perlin noise is unrealistically extreme (higher magnitude ratios and lower octaves), the code fails to produce meaningful outputs as indicated by missing data points in Fig S1_24 in S1 Appendix.

In Fig 4ci–4civ, we show the error on the column-column direction (i.e., horizontal) Green-Lagrange strain ($E_{cc}$) per subdomain for beat 4. We refer to this direction as the "column-column" direction to be consistent with the row (vertical) and column (horizontal) directions defined by our input images. Here, the reported $R^2$ values reveal that the minimum value of 0.943 occurs in subdomain B1. And, in general, these plots indicate that strain magnitudes tend to be underestimated. Furthermore, errors on $E_{cc}$ follow the observed trend for displacement errors, where synthetic examples of *sub*-pixel displacements exhibit higher errors with some cases having negative $R^2$ values (Tables S1_2 and S1_3 and Fig S1_8 in S1 Appendix).

In S1 Appendix, we share the validation results for all synthetic examples. We note that strain outputs are sensitive to subdomain divisions, specifically subdomain size. Determining a suitable subdomain size is a delicate process that is governed by two main opposing factors. The subdomain size should be large enough such that each subdivision contains an appropriate number of automatically identified fiducial markers to ensure that the computations are less sensitive to noise. On the contrary, the subdomain size should be small enough to avoid the loss or reduction of information due to averaging the heterogeneous deformation, or put more explicitly, obtaining attenuated or zero strain values due to lumping regions that are experiencing opposing deformations, for example extension versus compression. In general, based on our comprehensive experience with implementing the code on a number of synthetic and real examples, we recommend a subdomain side length that is between 30 and 40 pixels. From a methodological perspective, we propose that the user observes the displacement field and avoids having subdomains that span regions where the change in displacement flips sign. We recommend that manual examination of subdomain size be carried out once per dataset, for example, select a single movie from a batch of 100 to confirm that the subdomain division is appropriate. In the future, we plan to investigate the approach adopted in [76, 77] where strains are directly informed from affine warping functions optimized via the Lucas-Kanade algorithm [75] without computing displacement fields.

As described in this Section and in S1 Appendix, we have performed extensive validation studies for the "MicroBundleCompute" computational framework against a total of 416 synthetic examples of "Type 1" (16 examples generated under baseline conditions and the remaining 400 created by adversely altering the original 16 examples via the addition of Perlin noise)

and a single synthetic example of "Type 2" with known ground truth. These validation studies: 1) corroborate the output of the software against FEA-labelled data, 2) test its performance against realistic synthetic examples, and 3) evaluate its robustness against challenging examples with excessive noise artifacts. The obtained results reveal that for *super*-pixel displacements, our software is quite robust to all tested cases with relatively low magnitudes and high octaves, or in other words, when the Perlin noise patterns resemble speckle noise rather than pronounced textures (see Fig S1_4 in S1 Appendix). However, for examples with entirely *sub*-pixel displacements, the performance of the software degrades. In summary, "MicroBundle-Compute" breaks down when the synthetic data has noise artifacts that appear similar to the original texture, and for *sub*-pixel displacements. Yet, given that real experimental examples of beating microbundles generally produce displacements that exceed a single pixel and that Perlin noise examples with higher octaves more faithfully represent naturally occurring noise in real microbundle data than lower values, we anticipate that "MicroBundleCompute" will output reliable mechanical metrics in real experimental settings on condition that the natural contrast of the microbundle textures is visibly present and in focus in the time-lapse videos.

## Experimental data examples

We provide here a summary of implementing "MicroBundleCompute" on the experimental dataset described in the "Experimental data" Section and S2 Appendix, and shared on Dryad (https://doi.org/10.5061/dryad.5x69p8d8g) under a Creative Commons CC0 1.0 Universal Public Domain Dedication. Specifically, the "Microbundle Time-lapse Dataset" [96] contains all raw videos in ".tif" format for the 24 experimental time-lapse images of beating cardiac microbundles, 23 of which are brightfield videos, while the remaining single example is a phase contrast video. Besides the raw videos and the experimental metadata describing the conditions under which they were obtained, we include the tissue mask used for each example, whether generated automatically via our computational pipeline or manually via tracing in ImageJ [78]. These time-lapse videos and masks were used to generate the results shown here and in S2 Appendix. We note briefly that it is only possible to develop automatic mask segmentation functions for examples where there is imaging consistency and when we have an ample number of examples to identify a pattern. Future extensions of this framework will include automatic mask functions tailored to specific experimental needs.

In Fig 5, we show visualizations of output results generated by running "MicroBundleCompute" on 3 experimental examples, each from a different data type. Of note, we only visualize 3 of many potential software outputs: (1) full-field absolute displacement, (2) spatially distributed subdomain-averaged Green-Lagrange strain $E_{cc}$ (automatically rotated to align with the column-column horizontal direction and plotted at beat 0 strain peak), and (3) time series plot of $E_{cc}$ strains with respect to beat 0 frames. These are representative examples from the comprehensive list of outputs described in the software GitHub repository (https://github.com/HibaKob/MicroBundleCompute). While displacement and strain visualizations reveal spatial information on tissue heterogeneous behavior and spatial contraction patterns, time series strain plots highlight this spatial synchrony (or lack of synchrony) of subdomain contraction across each beat.

To complement the results in Fig 5, we show, in S2 Appendix, representative results obtained for all 24 real examples following the same format. Critically, all of the results presented in the supplementary document, for all 3 of the data types, were obtained without the need for any parameter tuning. As is evident in Fig S2_1 in S2 Appendix, we can group data from "Type 1" into 2 different categories: 1) examples 1–6 which constitute relatively challenging examples to the code since the imaging brightness/contrast is not optimized to fully

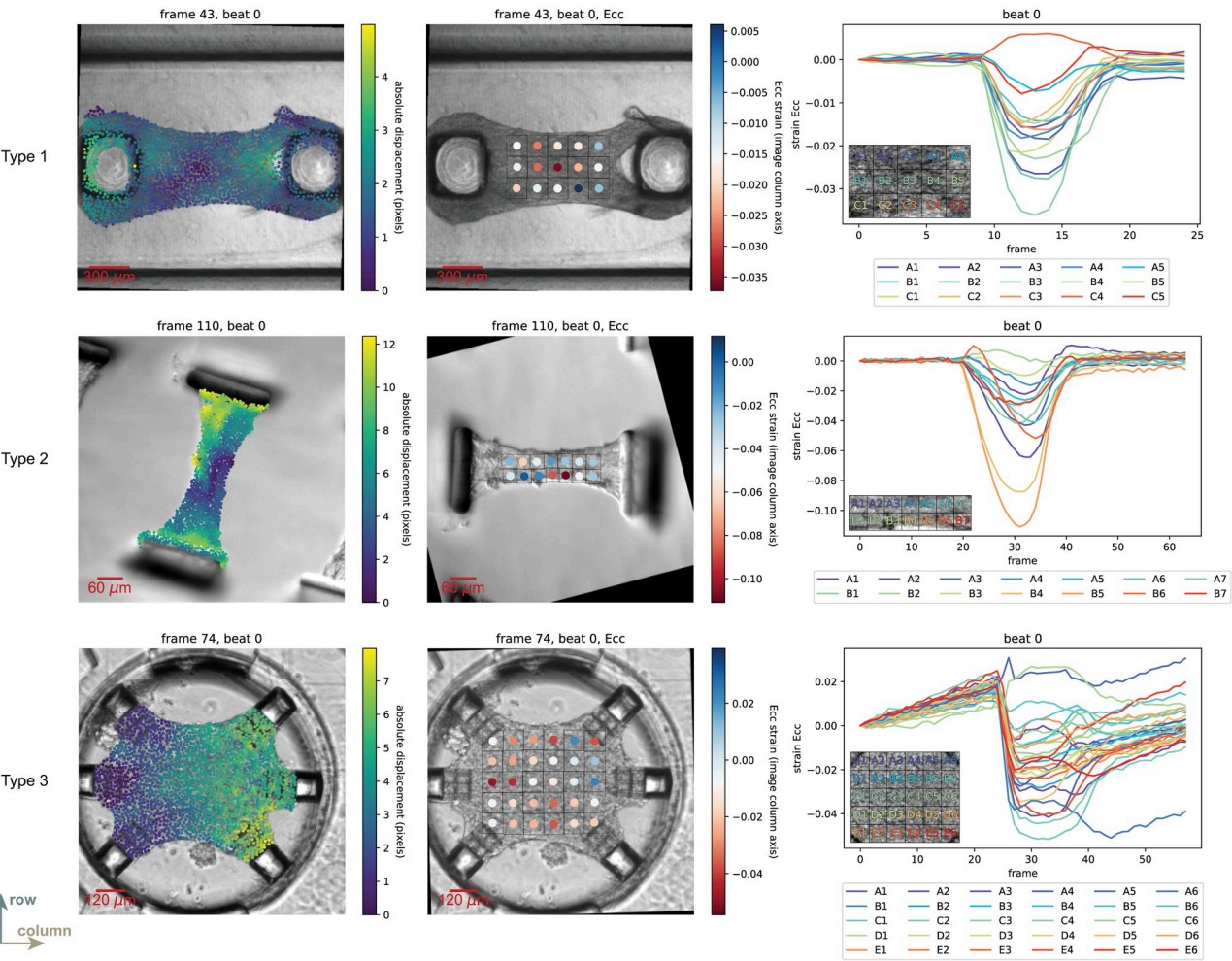

**Fig 5. Examples of "MicroBundleCompute" outputs.** We show sample outputs of the code when run on 3 examples from the 3 different data types, specifically Example 9 from "Type 1," Example 5 from "Type 2," and Example 1 from "Type 3" as shared on Dryad [96]. Note that the strain output for "Type 2" data is automatically rotated such that $E_{cc}$ aligns with the column-column direction. Additionally, the "Type 3" example shows an actuated microbundle at 0.5 Hz by applying sawtooth pressure waves with $\sim -6$ kPa peak amplitude (equivalent to $\sim 2.5\%$ strain) on the right side using a microfluidic pump (Elveflow OB1), and hence, the considerable discrepancy of the $E_{cc}$ time series plot from 0 at the end of beat 0.

accentuate the natural tissue texture, and 2) examples 7–11 where a texture is clearly visible in the microbundle region, making them a set of optimal examples for running the code. Despite the fact that "Type 1" data comprises the most consistent and well studied experimental platform, the full-field absolute displacement plots reveal wide variation in spatially distributed behavior. However, magnitude ranges are relatively consistent across samples with mean absolute displacement values at the peaks varying between 1.2 and 2.3 pixels (4.8 and 9.2 $\mu$m). We anticipate that this observation might be attributed to variations in pillar-tissue attachment behavior across different examples (see Fig S3_1 in S3 Appendix). However, further investigations are required to develop a systematic way to assess the effect of this pillar-tissue interaction on microbundle contractility and analyze the extracted metrics in this context.

Similar spatial heterogeneity is observed for the "Type 2" samples (Fig S2_2 in S2 Appendix), where examples 1–5, which were prepared under the same experimental conditions (soft, aligned matrix), exhibit discrepant displacement distributions but fairly consistent magnitude

ranges (mean absolute displacement values at the maximum contraction vary between 5.0 and 7.9 pixels or equivalently, 4.54 and 7.17 $\mu$m). For examples 6 and 7, which are prepared with stiff matrices, the results reveal lower contraction magnitudes where an aligned stiff matrix (example 6) shows higher contractions than the randomly distributed one (example 7) [94]. Of note, by examining all results of cardiac microbundle "Type 1" and "Type 2" data, a noticeable diagonal contraction pattern is apparent, especially in examples 2, 7, 9, and 10 from "Type 1" (Fig S2_1 in S2 Appendix) and examples 2, 3, and 5 from "Type 2" (Fig S2_2 in S2 Appendix). This observation, enabled by full-field tracking, indicates that future study to investigate the association between microbundle contraction, fiber alignment, and emergent load paths between the pillars, would be meaningful future work.

The diversity of "Type 3" experimental data prevents direct comparison between samples. However, the time series strain visualizations (Fig S2_3 in S2 Appendix) reveal that cardiac microbundles grown on these experimental constructs typically contract more synchronously than microbundles of "Type 1" and "Type 2," where the strain time series plots in Figs S2_1 and S2_2 in S2 Appendix respectively, show aspects of temporal heterogeneity for which peak contractions do not occur at the same frame within all subdomains per beat. Finally, the visualized average subdomain $E_{cc}$ strains, as well as the remaining two strain components (row-row direction Green-Lagrange strain $E_{rr}$ and column-row direction Green-Lagrange strain $E_{cr}$) that are computed and saved but not included in the set of representative outputs that we visualize here, give insight about the regions within the microbundle that are contracting or bulging in a given direction across each beat.

As we mention in the "Note on pillar tracking" Section and describe in more detail in S3 Appendix, it is straightforward to implement our computational framework to track pillar displacements, adding to the versatility of the "MicroBundleCompute" software framework. In Fig 6, we show, side by side, the outputs obtained via pillar tracking (Fig 6a-i and 6b-i) and via tracking the entire tissue domain (Fig 6a-ii, 6a-iii and 6b-ii, 6b-iii) for an example from "Type 1" and "Type 2" data respectively. Pillar tracking enables the calculation of an average absolute or directional pillar displacement based tissue force, which can be used to infer an average tissue stress given approximate tissue width and thickness as described in more detail in S3 Appendix. On the other hand, tissue tracking reveals abundant information regarding the inherent heterogeneous nature of microbundle beating. For example, full-field displacement fields (Fig 6a-ii and 6b-ii) show regions where maximum or minimum tissue contractions are taking place. Furthermore, while subdomain-averaged strains also underline the spatial heterogeneity of the tissue contractions, visualizing them with respect to time (or frame number) reveals the nature of temporal synchrony across the different subdivisions of the beating microbundle (Fig 6a-iii and 6b-iii). We note that a similar comparison can be carried out for all "Type 1" and "Type 2" data for which pillar tracking results are included in S3 Appendix, while tissue tracking results are shared in S2 Appendix.

Finally, we include in Fig 7 an example of "Type 3" data which clearly indicates that the microbundle is experiencing positive $E_{cc}$ strains at the center, specifically in subdomains A2 and A3 as indicated in the inset legend on the lower left corner of the strain time series plot. According to Wang et al. [107], this observation suggests that a necking instability is forming on this tissue over time, which is also supported by the thinning tissue width at the center. Within this context, being able to extract and examine strain distributions and magnitudes enables further investigations into understanding and determining the factors, such as pillar stiffness and ECM density [107], that produce microbundles that are more stable against necking.

Based on the results shown in this Section and in S2 Appendix, conventional metrics comprising tissue force and tissue stress obtained via basic pillar tracking offer insightful yet

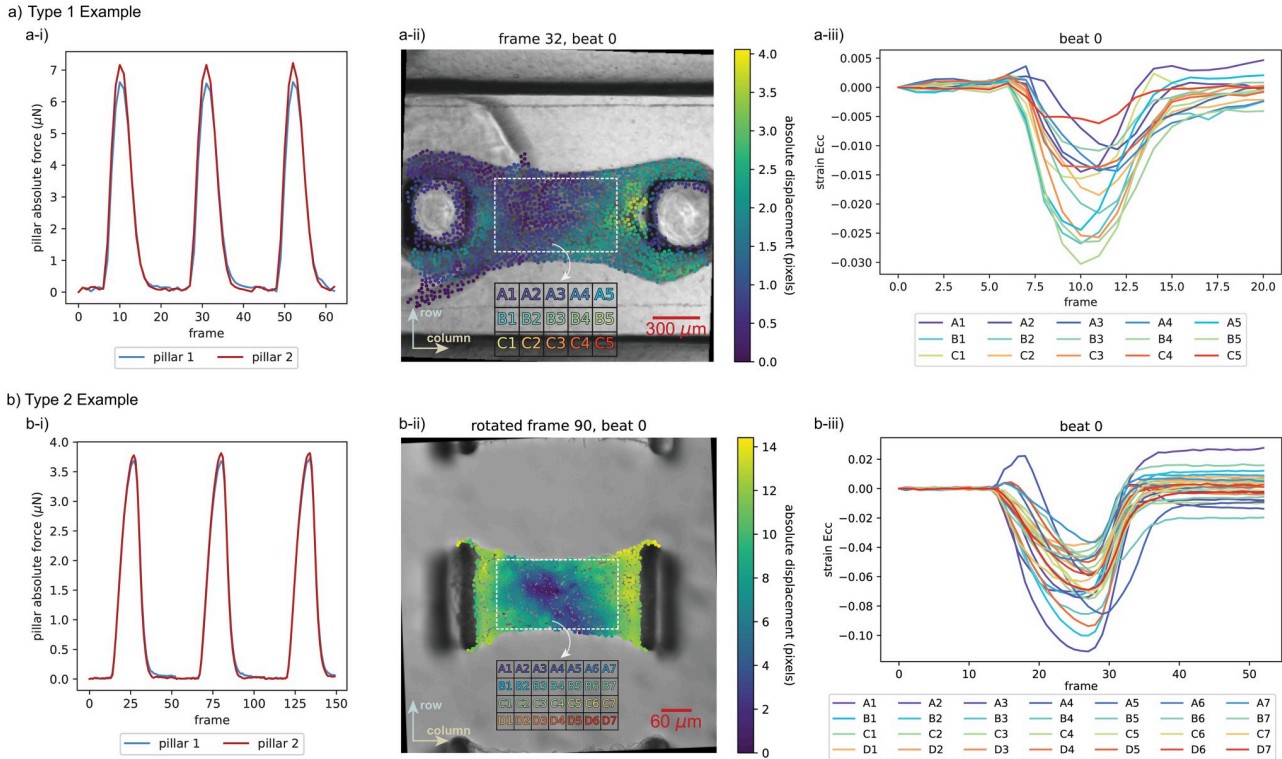

**Fig 6. Direct comparison of pillar tracking and tissue tracking.** Implementing the pillar tracking functionality of "MicroBundleCompute" on (a) Example 10 from "Type 1" data, (a-i) we obtain the pillar displacement based tissue force to describe the microbundle beating behavior, whereas full-field tissue tracking reveals the heterogeneous (a-ii) full displacement field as well as (a-iii) subdomain-averaged strains computed within the region marked by the dashed white box in (a-ii) where marker points within this region only are considered. In (b) we show the same outputs for Example 3 from "Type 2." We note that, for this example, we show the rotated displacement output to be consistent with the subdomain segmentation orientation. To view the original non-rotated displacement results, refer to Fig S2_2 in S2 Appendix.

lumped information about the microbundle behavior. Complemented with reliable and reproducible full-field data, such as displacement distributions and subdomain-averaged strains, as in [53] for example, these metrics become more useful to assessing the highly heterogeneous cardiac tissue behavior and understanding the complex underlying mechanisms driving this behavior. Furthermore, this spatial information would allow us to study injury

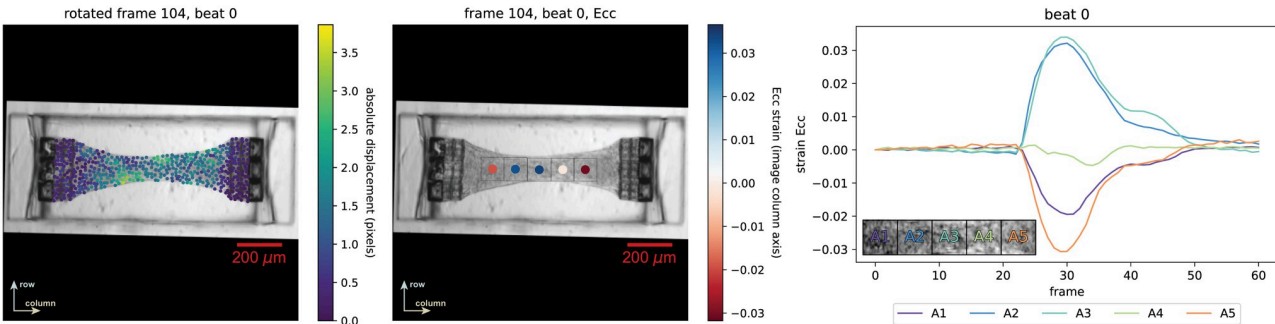

**Fig 7. Outputs of "MicroBundleCompute" run on Example 6 from "Type 3" reveal regions of tissue extension at the center.** We briefly note that for displacement outputs, rotating the results is optional, while for strain outputs, the frames are rotated such that the major axis of the microbundle automatically aligns with the column (i.e., horizontal) axis. To view the original orientation of this example, refer to Fig S2_3 in S2 Appendix.

models of tissue [53], as well as further investigate mechanical observations from pillar tracking, for example, the unbalanced pillar forces noted in Figs S3_1 and S3_2 in S3 Appendix. Finally, with sufficient collected data, this approach would enable the development of appropriate statistical models of the whole tissue. These endeavors are planned as part of our future work.

## Conclusion

In this work, we describe our approach to creating a computational tool for analyzing brightfield and phase contrast movies of beating microbundles. In brief, we describe our process for converting movies of beating microbundles to meaningful quantitative metrics, validating our approach against synthetically generated data, and testing it on a diverse pool of real experimental examples. In addition to ease of use, limited requirements for user intervention, relatively short run time, and no parameter tuning, "MicroBundleCompute" is easy to implement out of the box on new experimental datasets. Because it relies solely on the natural contrast of brightfield and/or phase contrast microbundle images and the resulting intensity gradients, it is also straightforward to integrate with existing experimental workflows. To enable broad adoption, we share the software under open-source license and look forward to receiving feedback from different users on how to adapt the code to tailor to their specific needs and enhance the overall user experience.

Looking forward, we aim to constantly improve the code. Future extensions will include automated quality control and pre-processing of input data, as well as enhancements to current functionality such as automatic adjustment of input movies that do not start from a fully relaxed frame. As viable alternatives arise, we also plan to benchmark it against available cardiac microbundle tissue analysis tools. In addition, we plan to continue developing our pipeline to address alternative quantitative metrics and different imaging modalities such as calcium imaging [108] and integrate these results with structural information such as sarcomere geometry and alignment [101, 109]. From the results shown in the "Experimental data examples" Section as well as S2 and S3 Appendices, it is also clear that there is significant variation across individual microbundle behavior both within and across testbeds. One key motivation for applying the "MicroBundleCompute" framework to these data moving forward is that it will make it possible to better understand and analyze this heterogeneity. In addition, extracting comparable mechanical metrics reliably and reproducibly across different testbeds allows for the identification of the favorable configurations and conditions that promote hiPSC-CM based tissue maturation, and ultimately, converge to an optimum system. Overall, our intention is for other researchers to directly benefit from disseminating this work. As a final note, here we demonstrate the utility and functionality of "MicroBundleCompute" for a particular highly used engineered cardiac tissue format: cardiac microbundles. In concurrent work [110], we leverage the fundamental core of this computational framework and make some minor modifications and extensions to extract mechanical metrics from actuated 2D muscle sheets. Looking forward, we will continue to generalize our framework to provide non-invasive, label-free, and high-throughput tools to facilitate contraction measurements across different engineered contractile tissue platforms to benefit the tissue engineering research community.

## Supporting information

**S1 Appendix. Additional details on pipeline validation against synthetic data.** Generating synthetic examples of both "Type 1" and "Type 2" as well as validating "MicroBundleCompute" against these synthetic data in their original form and with added Perlin noise are

described and discussed in more detail. **Table S1_1. Summary of validating tracked displacement against synthetic data of "Type 1" without any added Perlin noise.** Tracked mean absolute displacement is compared against a known ground truth. **Table S1_2. Summary of validating $E_{cc}$ strain against synthetic data of "Type 1" without any added Perlin noise.** Computed $E_{cc}$ strain from tracked displacement data is compared against a known ground truth. **Table S1_3. Tabulated summary of validating $E_{cc}$ strain outputs of "MicroBundle-Compute" against all synthetic data of "Type 1" without any added Perlin noise.** Computed $E_{cc}$ strain from tracked displacement data is compared against a known ground truth. **Table S1_4. Summary of validating tracked displacement against synthetic data of "Type 1" based on homogeneous activation with added Perlin noise.** Tracked mean absolute displacement is compared against a known ground truth and the "best" and "worst" results are reported. **Table S1_5. Summary of validating $E_{cc}$ strain from tracked displacement data against synthetic data of "Type 1" based on homogeneous activation with added Perlin noise.** Computed $E_{cc}$ strain from tracked displacement data is compared against a known ground truth and the "best," "average," and "worst" findings are reported. **Fig S1_1. Typical microbundle dimensions of "Type 1."** Schematic representation of the "Type 1" microbundle mesh implemented in our Finite Element simulations. **Fig S1_2. Convergence study of the grid size used to warp the microbundle textures to generate synthetic data.** Both $x$ and $y$ positions of the mesh cell centers are compared to a ground truth with respect to grid size. **Fig S1_3. A tabulated summary of the implemented conditions to obtain 400 noisy synthetic examples.** Main differences between the generated synthetic examples lie in the type of activation (homogeneous versus heterogeneous) used within the FEA model and the specific parameters of the added Perlin noise. **Fig S1_4. The effect of adding Perlin noise of different magnitude ratios and number of octaves on the tracked output of "MicroBundleCompute."** Real data contains inherent noise that a range of Perlin noise parameters can mimic its effects. **Fig S1_5. Displacement validation against examples based on homogeneous activation.** A detailed comparison is shown between tracked mean absolute displacement and a known ground truth for 8 different synthetic examples. **Fig S1_6. Displacement validation against examples based on heterogeneous activation.** A detailed comparison is shown between tracked mean absolute displacement and a known ground truth for 8 different synthetic examples. **Figs S1_7–S1_22. Detailed validation results of $E_{cc}$ strain for all 16 synthetic examples without the addition of Perlin noise.** Computed $E_{cc}$ strain from tracked displacements are compared to a known ground truth within each subdomain. **Fig S1_23. Effect of adding Perlin noise to synthetic data of "Type 1" based on FE simulations with homogeneous activation.** Perlin noise generated for 5 different magnitude ratios and 5 different octaves are added to synthetic data to test the performance of "MicroBundleCompute" in tracking displacements that exceed a single pixel. **Fig S1_24. Effect of adding Perlin noise to synthetic data of "Type 1" based on FE simulations with heterogeneous activation.** Perlin noise generated for 5 different magnitude ratios and 5 different octaves are added to synthetic data to test the performance of "MicroBundleCompute" in tracking *sub*-pixel displacements. **Fig S1_25. Displacement validation against synthetic data of "Type 2."** Tracked displacements in $X$ and $Y$ are compared to a known ground truth. **Fig S1_26. Strain validation against synthetic data of "Type 2."** Computed $E_{cc}$ strains based on tracked displacements are compared to a known ground truth. **Fig S1_27. Validation via manual tracking against "Type 2" data.** Selected points are manually tracked and the displacements in $X$ and $Y$ are compared to the equivalent outputs obtained by running "MicroBundleCompute". (PDF)

**S2 Appendix. Testing "MicroBundleCompute" with additional examples of real data.** The results of implementing "MicroBundleCompute" on a total of 24 experimental time-lapse images of cardiac microbundles, 11 examples of "Type 1," 7 of "Type 2," and 6 of "Type 3." **Table S2_1. A summary of the experimental conditions associated with each example movie.** The details of experimental conditions include image acquisition parameters as well as pillar stiffness values for "Type 1" and "Type 2" data. **Table S2_2. Additional details for each example of "Type 1" data.** These details include example-specific information, code implementation details, and subdomain segmentation parameters. **Table S2_3. Additional details for each example of "Type 2" data.** These details include example-specific information, code implementation details, and subdomain segmentation parameters. **Table S2_4. Additional details for each example of "Type 3" data.** These details include example-specific information and subdomain segmentation parameters. **Fig S2_1. Example outputs of "MicroBundleCompute" run on "Type 1" experimental data.** The provided outputs include full-field mean absolute displacement and subdomain-averaged Green-Lagrange $E_{cc}$ strain at the first tracked peak, as well as a time series plot of $E_{cc}$ strain for the first tracked beat. **Fig S2_2. Example outputs of "MicroBundleCompute" run on "Type 2" experimental data.** The provided outputs include full-field mean absolute displacement and subdomain-averaged Green-Lagrange $E_{cc}$ strain at the first tracked peak, as well as a time series plot of $E_{cc}$ strain for the first tracked beat. **Fig S2_3. Example outputs of "MicroBundleCompute" run on "Type 3" experimental data.** The provided outputs include full-field mean absolute displacement and subdomain-averaged Green-Lagrange $E_{cc}$ strain at the first tracked peak, as well as a time series plot of $E_{cc}$ strain for the first tracked beat.
(PDF)

**S3 Appendix. Additional details on pillar tracking.** The implementation of the pillar tracking pipeline within "MicroBundleCompute" is explained in more details and demonstrated on 11 examples of "Type 1" and 7 examples of "Type 2." **Fig S3_1. Pillar tracking on "Type 1" examples.** Depicted results include pillar absolute force ($\mu$N) obtained on both pillars. **Fig S3_2. Pillar tracking on "Type 2" examples.** Depicted results include pillar absolute force ($\mu$N) obtained on both pillars.
(PDF)

**S1 Movie. Synthetic example based on "Type 2" data.** This movie provides the synthetic example generated based on "Type 2" data as described in S1 Appendix.
(TIF)

**S2 Movie. Movie of tracked absolute displacement for Example 9 from "Type 1" data.** This movie is provided as a supplement to Fig S2_1, Example 9, in S2 Appendix.
(MP4)

**S3 Movie. Movie of tracked $E_{cc}$ strain for Example 9 from "Type 1" data.** This movie is provided as a supplement to Fig S2_1, Example 9, in S2 Appendix.
(MP4)

**S4 Movie. Movie of tracked absolute displacement for Example 10 from "Type 1" data.** This movie is provided as a supplement to Fig S2_1, Example 10, in S2 Appendix.
(MP4)

**S5 Movie. Movie of tracked $E_{cc}$ strain for Example 10 from "Type 1" data.** This movie is provided as a supplement to Fig S2_1, Example 10, in S2 Appendix.
(MP4)

**S6 Movie. Movie of tracked absolute displacement for Example 3 from "Type 2" data.** This movie is provided as a supplement to Fig S2_2, Example 3, in S2 Appendix.
(MP4)

**S7 Movie. Movie of tracked $E_{cc}$ strain for Example 3 from "Type 2" data.** This movie is provided as a supplement to Fig S2_2, Example 3, in S2 Appendix.
(MP4)

**S8 Movie. Movie of tracked absolute displacement for Example 5 from "Type 2" data.** This movie is provided as a supplement to Fig S2_2, Example 5, in S2 Appendix.
(MP4)

**S9 Movie. Movie of tracked $E_{cc}$ strain for Example 5 from "Type 2" data.** This movie is provided as a supplement to Fig S2_2, Example 5, in S2 Appendix.
(MP4)

**S10 Movie. Movie of tracked absolute displacement for Example 1 from "Type 3" data.** This movie is provided as a supplement to Fig S2_3, Example 1, in S2 Appendix.
(MP4)

**S11 Movie. Movie of tracked $E_{cc}$ strain for Example 1 from "Type 3" data.** This movie is provided as a supplement to Fig S2_3, Example 1, in S2 Appendix.
(MP4)

**S12 Movie. Movie of tracked absolute displacement for Example 6 from "Type 3" data.** This movie is provided as a supplement to Fig S2_3, Example 6, in S2 Appendix.
(MP4)

**S13 Movie. Movie of tracked $E_{cc}$ strain for Example 6 from "Type 3" data.** This movie is provided as a supplement to Fig S2_3, Example 6, in S2 Appendix.
(MP4)

## Acknowledgments

We gratefully acknowledge the collaborative opportunities facilitated by the CELL-MET Engineering Research Center. We especially thank the administrative team at CELL-MET for coordinating the Research Experiences for Undergraduates (REU) program. We would also like to acknowledge the staff at Boston University libraries for providing advice regarding data dissemination practices.

## Author Contributions

**Conceptualization:** Hiba Kobeissi, Emma Lejeune.

**Data curation:** Hiba Kobeissi, Javiera Jilberto, M. Çağatay Karakan, Xining Gao, Samuel J. DePalma, Shoshana L. Das, Lani Quach, Emma Lejeune.

**Formal analysis:** Hiba Kobeissi, Emma Lejeune.

**Funding acquisition:** Brendon M. Baker, Christopher S. Chen, David Nordsletten, Emma Lejeune.

**Investigation:** Hiba Kobeissi, Emma Lejeune.

**Methodology:** Hiba Kobeissi, Javiera Jilberto, M. Çağatay Karakan, Xining Gao, Samuel J. DePalma, Shoshana L. Das, Emma Lejeune.

**Project administration:** Hiba Kobeissi, Emma Lejeune.

**Resources:** Brendon M. Baker, Christopher S. Chen, David Nordsletten, Emma Lejeune.

**Software:** Hiba Kobeissi, Javiera Jilberto, Jonathan Urquia, Emma Lejeune.

**Supervision:** Brendon M. Baker, Christopher S. Chen, David Nordsletten, Emma Lejeune.

**Validation:** Hiba Kobeissi, Javiera Jilberto, Emma Lejeune.

**Visualization:** Hiba Kobeissi, Javiera Jilberto, Emma Lejeune.

**Writing – original draft:** Hiba Kobeissi, Javiera Jilberto, M. Çağatay Karakan, Xining Gao, Samuel J. DePalma, Shoshana L. Das, David Nordsletten, Emma Lejeune.

**Writing – review & editing:** Hiba Kobeissi, Brendon M. Baker, Christopher S. Chen, Emma Lejeune.

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
