## [Decision Letter · Decision Letter 0]

9 Oct 2023

PONE-D-23-28138MicroBundleCompute: Automated segmentation, tracking, and analysis of subdomain deformation in cardiac microbundlesPLOS ONE

Dear Dr. Lejeune,

Thank you for submitting your manuscript to PLOS ONE. After careful consideration, we feel that it has merit but does not fully meet PLOS ONE’s publication criteria as it currently stands. Therefore, we invite you to submit a revised version of the manuscript that addresses the points raised during the review process.

We look forward to receiving your revised manuscript.

Kind regards,

Mohammad Amin Fraiwan

Academic Editor

PLOS ONE

 [This work was supported by the CELL-MET Engineering Research Center National Science Foundation ECC-1647837. See:https://www.bu.edu/cell- met/andhttps://nsf.gov/awardsearch/showAward?AWD_ID=1647837].  

5. We note that Figure 1, 2, 3, 4A, 5, 6, 7, S1_1, S1_7, S1_8, S1_9, S1_10, S1_11, S1_12, S1_13, S1_14, S1_15, S1_16, S1_17, S1_18, S1_19, S1_20, S1_21, S1_22, S1_25, S1_26, S2_1, S2_2, S2_3, S3_1, S3_2 and  S1 to S13. in your submission contain copyrighted images. All PLOS content is published under the Creative Commons Attribution License (CC BY 4.0), which means that the manuscript, images, and Supporting Information files will be freely available online, and any third party is permitted to access, download, copy, distribute, and use these materials in any way, even commercially, with proper attribution. For more information, see our copyright guidelines: http://journals.plos.org/plosone/s/licenses-and-copyright.

A. You may seek permission from the original copyright holder of Figure 1, 2, 3, 4A, 5, 6, 7, S1_1, S1_7, S1_8, S1_9, S1_10, S1_11, S1_12, S1_13, S1_14, S1_15, S1_16, S1_17, S1_18, S1_19, S1_20, S1_21, S1_22, S1_25, S1_26, S2_1, S2_2, S2_3, S3_1, S3_2 and  S1 to S13 to publish the content specifically under the CC BY 4.0 license. 

B. If you are unable to obtain permission from the original copyright holder to publish these figures under the CC BY 4.0 license or if the copyright holder’s requirements are incompatible with the CC BY 4.0 license, please either i) remove the figure or ii) supply a replacement figure that complies with the CC BY 4.0 license. Please check copyright information on all replacement figures and update the figure caption with source information. If applicable, please specify in the figure caption text when a figure is similar but not identical to the original image and is therefore for illustrative purposes only.

7. We notice that your supplementary [figures/tables] are included in the manuscript file. Please remove them and upload them with the file type 'Supporting Information'. Please ensure that each Supporting Information file has a legend listed in the manuscript after the references list.

8. We are unable to open your Supporting Information file [_MACOSX]. Please kindly revise as necessary and re-upload.

Reviewers' comments:

Reviewer's Responses to Questions

**Comments to the Author**

1. Is the manuscript technically sound, and do the data support the conclusions?

Reviewer #1: Yes

Reviewer #2: Yes

Reviewer #3: Yes

2. Has the statistical analysis been performed appropriately and rigorously? 

Reviewer #1: Yes

Reviewer #2: Yes

Reviewer #3: Yes

3. Have the authors made all data underlying the findings in their manuscript fully available?

Reviewer #1: Yes

Reviewer #2: Yes

Reviewer #3: Yes

4. Is the manuscript presented in an intelligible fashion and written in standard English?

Reviewer #1: Yes

Reviewer #2: Yes

Reviewer #3: Yes

5. Review Comments to the Author

Reviewer #1: This work describes an open-source Python tool, MicroBundleCompute that computationally tracks the deformation from movie data of in vitro hiPSC-CM and predicts the physical properties such as strain. The tool is validated on synthetic data from FEA simulations mimicing the experimental setup. It is also demonstrated on various experimental data. The manuscript is well written with detailed explanations and many examples. The software is also well implemented with clear tutorials. While the methodological novelty is rather limited, I believe this work provides a versatile and reliable computational tool for high-throughput evaluation of hiPSC-CM experimental data. I only have some minor comments.

1.It would be helpful to add validation results on synthetic data for type 2 and type 3 as well as done for type 1 in Figure 4.

2.Two other “less automated” methods were mentioned in Introduction, “Huebsch, Nathaniel, et al. "Automated video-based analysis of contractility and calcium flux in human-induced pluripotent stem cell-derived cardiomyocytes cultured over different spatial scales." Tissue Engineering Part C: Methods 21.5 (2015): 467-479.” and “Méry, Adrien, et al. "Light-driven biological actuators to probe the rheology of 3D microtissues." Nature Communications 14.1 (2023): 717.” Some comparison should be performed with these methods on both synthetic data and experimental data.

Reviewer #2: The authors present a method for quantifying the contraction of cardiomyocyte microbundles, a problem which is currently very topical, with the recent rise of engineered heart tissue studies. The authors further present a method for generating simulated videos of contracting microbundles with solid modelling background. The paper is well written and the methods thoroughly documented. With the software released as modular and open source software, it is my belief that the paper and software will prove to be very useful for researchers working in the field. The language is impeccable.

In general, I only have few remarks regarding the manuscript.

1) The addition of Perlin noise for the simulated data is a good call. However, I am not convinced it fully recapitulates the noise sources stemming from the imaging process. The addition of shot noise could induce another source of error commonly present in the data. The authors could discuss if the way the Perlin noise is applied is sufficient to simulate this source of error.

2) The authors estimate the microbundle thickness to be a 400 um in the FEM simulation, instead of varying as a function of distance from the micropillars. I would like the authors to discuss the impact of the design decision on the model validity, or to provide a more detailed justification for the choice.

3) The impact of the current study may remain obscure to a reader if they're not well acquainted with the field. In particular, I feel the conclusion should more clearly reflect the advantages of cross-system compatibility how improvements in the field relate to tissue engineering studies.

4) While the thoroughly reported supplementary material is informative, in my opinion it could be more organized for easier reading. In its current form, it is difficult to follow and appreciate the 15 pages of error calculations (pages 44 to 59). I am not convinced this way of reporting the findings is necessary, or if e.g. the MAE in subdomains could simply be stated numerically for more compact presentation.

Reviewer #3: The manuscript titled “MicroBundleCompute: automated segmentation, tracking, and analysis of subdomain deformation in cardiac microbundles” presented an open-source computational tool able to analyze the beating of cardiac microbundles and provide quantification on their mechanical metrics. The authors performed an extensive validation on several experimental datasets and conclude that due to its ease of use and low computational demand this computational tool could be broadly used to analyze beating microbundles. The paper is well written, however, there are some major concerns to address:

1. Could the authors explain in more detail how the three-dimensional microbundle geometry is generated from the 2D coordinates? Authors refer to a reasonable tissue thickness of 400 um to extrude the 2D surface, but is that value is dependent on the different experimental dataset. Cross-sections of the tissues should be used instead to measure the tissue diameter and use that as input. Could the authors elaborate on how to adapt this parameter to each tissue type?

2. In page 13, authors model the cardiac tissue as a nearly-incompressible transversely isotropic hyperelastic material. How does this take into account different cell composition of the microbundles and the different ECM, which may change over time and depending on the cellular composition? Have the authors considered these implications?

3. In page 18, “By design, our computational framework requires a minimum of 3 beats to be present…”. How do the authors plan to account for experimental conditions where tissues decrease considerably their frequency of contraction? How long do the vides need to be?

4. In page 20, point 2. How do the authors guarantee that the tissues are in the relaxed position?

5. In page 24, “From a methodological perspective,…”. Does that mean that the user requires a specific training? The fact that a user has to observe the samples and choose the subdomain size per video suggests that each video has to be analyzed manually and not automatically, which hinders the high-throughput claim of the software. Can the authors elaborate on this?

6. In page 25, “time series strain plots highlight the synchrony {or lack of synchrony)…”. Can the authors elaborate on this point? Where the tissues electrically paced? What is the lowest and the highest frequency that the software can track?

7. Force measurements from pillar tracking are not well described per type of experimental set up. How do the authors plan on implementing this functionality according to each type of microbundle that the users might have (including different geometry of the pillars and young modulus)?

8. In figure 6a-II there are some purple dots outside of the tissue region. How does the author exclude these data or avoid to take into account artifacts in the strain calculations?

Minor concerns:

1. In abstract: “requires little to no parameter tuning” is a misleading sentence that should be adjusted.

2. In abstract: “runs quick on a personal computer”. Compared to what? Should be rephrased to “low computational demand”

3. It is not clear if the software has an UI to benefit the scientific community, can the authors elaborate on that? Besides, is the software compatible only with windows or can it work on macOS and linux?

4. In page 12 “Overall, these approaches are suitable when only basic information…”. Previous sentence mentioned software also provide absolute values of force. This sentence may be misleading to the scientific community.

5. In page 14 and 15, authors refer to recording of time-lapse videos of tissue contractions at 30Hz or 65Hz. Those are very low frame rates to record the tissues. Can the authors explain these choices?

6. In tracking region mask generation. It is not clear if the user will have to do it manually or if it is automatically generated by the software. In the case that is generated by the software, which is the criteria to select between the 3 segmentation methods? Which one is the most accurate method to create the mask?

7. In page 24, “output reliable mechanical metrics in real experimental settings on condition….”. Is the software able to give as an output the noise level of the input sample and show the user the internal modifications done by the software?

8. In page 24, “while the remaining single example is a phase contrast video”. Why did the authors choose phase contrast?

6. PLOS authors have the option to publish the peer review history of their article (what does this mean?). If published, this will include your full peer review and any attached files.

Reviewer #1: No

Reviewer #2: No

Reviewer #3: No

---

## [Author Response · Author response to Decision Letter 0]

28 Nov 2023

Please see our formatted "Response to Reviewers" document uploaded with the submission.

---

## [Decision Letter · Decision Letter 1]

1 Feb 2024

MicroBundleCompute: Automated segmentation, tracking, and analysis of subdomain deformation in cardiac microbundles

PONE-D-23-28138R1

Dear Dr. Lejeune,

We’re pleased to inform you that your manuscript has been judged scientifically suitable for publication and will be formally accepted for publication once it meets all outstanding technical requirements.

Kind regards,

Mohammad Amin Fraiwan

Academic Editor

PLOS ONE

Reviewers' comments:

Reviewer's Responses to Questions

**Comments to the Author**

1. If the authors have adequately addressed your comments raised in a previous round of review and you feel that this manuscript is now acceptable for publication, you may indicate that here to bypass the “Comments to the Author” section, enter your conflict of interest statement in the “Confidential to Editor” section, and submit your "Accept" recommendation.

Reviewer #1: All comments have been addressed

Reviewer #2: All comments have been addressed

Reviewer #3: All comments have been addressed

2. Is the manuscript technically sound, and do the data support the conclusions?

Reviewer #1: Yes

Reviewer #2: Yes

Reviewer #3: Yes

3. Has the statistical analysis been performed appropriately and rigorously? 

Reviewer #1: Yes

Reviewer #2: Yes

Reviewer #3: Yes

4. Have the authors made all data underlying the findings in their manuscript fully available?

Reviewer #1: Yes

Reviewer #2: Yes

Reviewer #3: Yes

5. Is the manuscript presented in an intelligible fashion and written in standard English?

Reviewer #1: Yes

Reviewer #2: Yes

Reviewer #3: Yes

6. Review Comments to the Author

Reviewer #1: (No Response)

Reviewer #2: The authors have answered my questions and comments in full, and provided additional experimental data to support their claims. I have to further questions or comments, and recommend accepting the manuscript.

Reviewer #3: The authors have adressed my concerns and significantly improved the content and readibility of the manuscript. I do agree that the field will benefit to have access to this tool.

7. PLOS authors have the option to publish the peer review history of their article (what does this mean?). If published, this will include your full peer review and any attached files.

Reviewer #1: No

Reviewer #2: No

Reviewer #3: No

---

## [Editor Report · Acceptance letter]

15 Mar 2024

PONE-D-23-28138R1 

PLOS ONE

Dear Dr. Lejeune, 

I'm pleased to inform you that your manuscript has been deemed suitable for publication in PLOS ONE. Congratulations! Your manuscript is now being handed over to our production team.

Kind regards, 

on behalf of

Dr. Mohammad Amin Fraiwan 

Academic Editor

PLOS ONE